# Incorporating the Benefits of Geosynthetic into MEPDG

Murad Abu-Farsakh [1,2,*], Mehdi Zadehmohamad [1] and George Z. Voyiadjis [1]

1    Department of Civil and Environmental Engineering, Louisiana State University, Baton Rouge, LA 70803, USA
2    Louisiana Transportation Research Center, Louisiana State University, 4101 Gourrier Avenue,
    Baton Rouge, LA 70808, USA
*    Correspondence: cefars@lsu.edu

**Abstract:** One of the most effective ways to increase the longevity of pavement structures is through the integration of geosynthetic reinforcement. Geosynthetics are synthetic materials such as geotextiles, geogrids, or geocomposites that are added to the interface between the subgrade and the base layer of a pavement structure. To evaluate the effect of various parameters on the structural benefits of geosynthetic reinforcement on the pavement structure of low-volume traffic flexible pavements, a finite element (FE) study was performed using the ABAQUS program. These parameters included the geosynthetic type, geosynthetic tensile stiffness, subgrade stiffness, and base thickness. The FE rutting curves for the 100 cycles were calibrated using the mechanistic–empirical (M-E) transfer functions, which were then used to calculate the long-term rutting curves. The traffic benefit ratio (TBR) was initially calculated based on the calibrated rutting curves for each pavement layer. The calculated TBRs were then used as an input in AASHTOWare to compute the base effective resilient modulus ($M_{R\text{-eff}}$) and the factor of base course reduction (BCR). The results showed that adding one layer of geosynthetics enhanced the rutting performance of pavement structures significantly (up to 8.9 in TBR, 322% in $M_{R\text{-eff}}$, and 64% in BCR). Geogrids showed higher benefits than geotextiles due to the interlocking between base aggregates and geogrid aperture. The values of TBR, $M_{R\text{-eff}}$, and BCR increase with the increasing tensile stiffness of the geosynthetics and the rutting target and with the decreasing subgrade stiffness. The results also demonstrated peak values of TBR, $M_{R\text{-eff}}$, and BCR for a base thickness of 25.4 cm.

**Keywords:** FEM; MEPDG; geosynthetic; TBR; $M_{R\text{-eff}}$; BCR

## 1. Introduction

Pavement failure refers to the loss of integrity or functional performance of a road surface. There are several different mechanisms that can contribute to pavement failure, including excessive loading from vehicles, weathering and exposure to environmental elements such as moisture and temperature changes, material fatigue due to repeated loading, and aging of the pavement structure. One of the most known pavement failure mechanisms (or distresses) is the rutting at the surface of the pavement structure, which can be a result of densification in pavement layers or deformation of weak natural subgrade soil or both [1]. In the state of Louisiana, the main problematic issue that is associated with different pavement distresses is the presence of weak natural soil. A common practice for treating weak subgrade soil in Louisiana is by stabilizing the surface layer with cement and/or lime, depending on the soil type. The increased stiffness of stabilized soil would act as a platform that causes the bottom unstabilized soil to receive a lower magnitude of the applied traffic and pavement loads, thus lowering the highest vertical pressures on top of the natural soil/subgrade. The stated method is not practical on very weak natural soils and can have environmental drawbacks. In addition, the required time for the construction sequence of stabilization can be a problem in many projects. Using geosynthetic reinforcement can provide an effective alternative solution for soil treatment/stabilization that can alleviate the aforementioned problems in the pavement structure. In addition,

geosynthetics can offer many economic benefits for pavement projects by increasing the speed of construction and extending the pavement's life expectancy in the long run.

Many different applications have made use of geosynthetic materials such as filtration, separation, and reinforcement in pavement construction. Many types of research have been carried out to calculate the advantages of utilizing various kinds of geosynthetic products in pavement construction [1–9]. Geogrids and geotextiles are the most utilized types of geosynthetic products in pavement construction, which have been evaluated in many previous research studies. The results of other research studies have shown that using geosynthetics to reinforce pavements has been proven to be an effective approach for decreasing the rutting depth of pavements [10–12]. As a result, the lifespan of a pavement can be increased significantly [13–15]. Additionally, for the same service life of the unreinforced section, the designed thickness of the base course layer can be minimized by utilizing the geosynthetic reinforcement in pavements [2,15,16]. Geosynthetic reinforcement allows pavement structures to be constructed on weak soils [17,18]. The geosynthetics' reinforcement performance in pavement structures is substantially influenced by the different sections and material characteristics such as asphalt and the base course thicknesses, the stiffness of natural/subgrade soil, geosynthetics type, stiffness, and location [19–23].

Despite the growing practical applications of geosynthetics in pavement construction, the application of design theories, design methodologies, and design guidelines for using geosynthetics in pavement structure construction is evolving at a pretty slow rate. There are several studies in the literature aimed at incorporating geosynthetics reinforcement in pavement section design by introducing new design methodologies and guidelines for different pavement requirements and conditions [3,5,13].

The framework of the AASHTO 1993 is largely utilized in the current ways of adding geosynthetic reinforcement in the design of flexible pavement structures. In this framework, the base course layer structural number is increased by a certain factor when the pavement section is reinforced with geosynthetic reinforcement. As a consequence, the design method causes a decrease in the base course layer thickness, when a geosynthetic material is used as a reinforcement [6,16]. The new Mechanistic–Empirical Pavement Design Guide (MEPDG) released in 2008 is not only founded on the empirical methods for pavement section design, the mechanistic part of the MEPDG includes the elastic responses of the pavement layers in the process of design, while the empirical part uses the elastic responses to derive the long-term distresses of the pavement section using empirical transfer functions [24,25]. The effect of geosynthetic reinforcement should be reflected in either the mechanistic or the empirical part of the MEPDG design. However, since there is a lack of knowledge on the geosynthetic reinforcement mechanisms within the MEPDG design process and there is an absence of enough studies to calculate the geosynthetic benefits for pavement structures, there is no nationally or universally accepted specification or approach for reinforced flexible pavement design. As a result, research into and studies on reinforced flexible pavements in the context of MEPDG are still ongoing.

## 2. Objectives

The current study aims to incorporate the structural benefits of geosynthetic reinforcement on flexible pavement structure rutting performance for low-volume traffic roads based on the MEPDG design concept. This study quantifies the use of thorough finite element analysis to examine the impact of numerous variables that contribute to the advantages of geosynthetic reinforcement in flexible pavement constructions and through a parametrical study to determine the long-term traffic benefit ratio (TBR) benefits of geosynthetic reinforcement. Using AASHTOWare 2.6.0 software, the outcomes of the FE models were combined with the mechanistic–empirical (ME) method for calculating the structural contribution of geosynthetics to reinforced pavement sections such as the base course effective resilient modulus ($M_{R\text{-eff}}$) and the base course reduction (BCR) factor.

## 3. Finite Element (FE) Modeling

The commercial finite element program ABAQUS was used in this study to explore the behavior of geosynthetic-reinforced pavement sections under repeated traffic loadings. For this purpose, several FE models were built and evaluated employing two-dimensional axisymmetric conditions to explore the effectiveness of various parameters in the rutting performance of geosynthetic-reinforced pavements. For all of the pavement sublayers, eight-node quadrilateral elements with biquadratic axisymmetric (CAX8R) were adopted for the element properties in the model. For the geosynthetic reinforcement in the models which is placed at the top of the subgrade layer (red dashed line in Figure 1), the adopted elements are three-node membranes. For all of the modeling, an extremely fine-meshed model with a total of 9176 elements was chosen (6572 elements for the subgrade, 1860 elements for the base, and 744 elements for HMA). The schematic view and dimensions of the finite element models used in this investigation are shown in Figure 1.

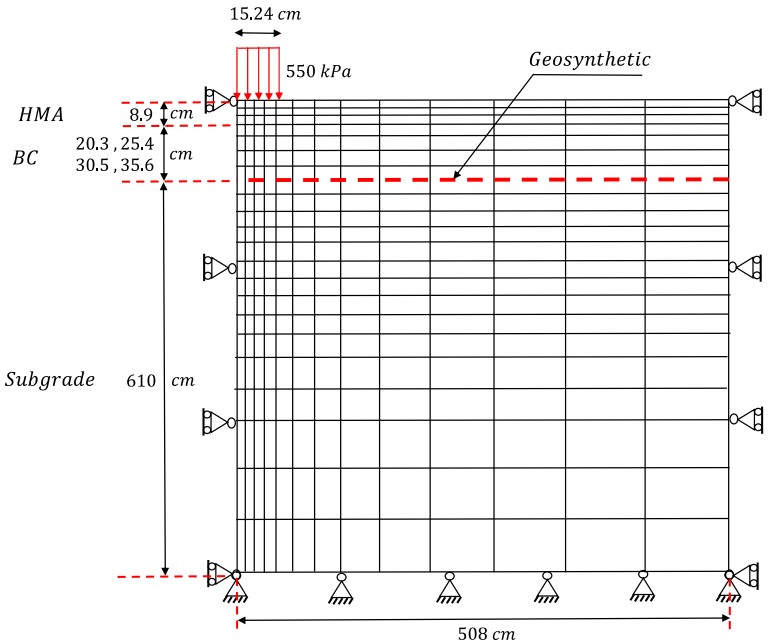

**Figure 1.** Schematic view of geosynthetics reinforcement at the base/subgrade interface and dimensions of the FE models.

In order to capture the realistic responses of the pavement sections under traffic loadings in finite element models, the following models were adopted to model the sublayers' behavior in the pavement sections.

### 3.1. Hot Mixture Asphalt (HMA)

In this study, time-dependent (viscoelastic) behavior was selected to simulate the behavior of HMA materials under repeated wheel loadings. In the first step, the shear and bulk modulus of the HMA material are required to be characterized to simulate the viscoelastic behavior of HMA. The Prony series can be used in the finite element numerical simulation to model the HMA viscoelastic behavior, which can be defined as the following equations (Equations (1) and (2)):

$$G(t) = G_0 \left( 1 - \sum_{i=1}^{n} G_i \left( 1 - e^{-t/\tau} \right) \right) \tag{1}$$

$$K(t) = K_0 \left( 1 - \sum_{i=1}^{n} K_i \left( 1 - e^{-t/\tau} \right) \right) \tag{2}$$

where $G(t)$ is the relaxation shear modulus, $K(t)$ is the relaxation bulk modulus, $G_0$ is the instantaneous shear modulus, $K_0$ is the instantaneous bulk modulus, and $G_i$, $K_i$, and $\tau$ are the input coefficients.

The Prony series should be fitted to the HMA dynamic modulus master curve for at least five stages to derive the Prony series parameters ($\tau$, $G$, and $k$) for use in the FE models. In this study, the properties of the HMA Prony series adopted from a previous study [21] was used here for the numerical simulation as presented in Table 1.

**Table 1.** HMA Prony series parameters.

| Elastic Properties | Poisson's Ratio | | | | 0.35 | |
|---|---|---|---|---|---|---|
| | Instantaneous Modulus (MPa) | | | | 3447 | |
| Viscoelastic Properties | $g_i$ , $k_i$ | 0.452 | 0.278 | 0.148 | 0.108 | 0.00746 | 0.00436 |
| | $\tau_i$ | 0.000113 | 0.00314 | 0.013 | 0.184 | 2.29 | 25.7 |

### 3.2. Unbound Granular Base Course

The behavior of the granular base course material was modeled using the modified drucker–prager with cap (MDPC) constitutive model in the FE study. The MDPC plasticity model has been used for numerical simulation by geotechnical researchers in various geotechnical problems, and it is capable of considering the hardening mechanism to account for plastic compaction. The yield surface of the MDPC constitutive model has three parts (Figure 2); the shear failure line, the elliptical cap, and the region of transition, which seamlessly joins the cap and shear failure surface.

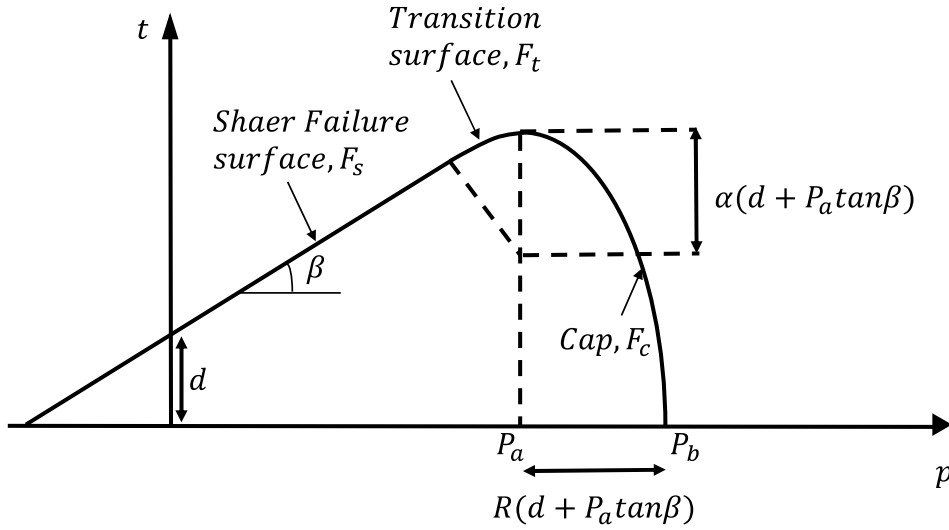

**Figure 2.** Yield surfaces of the MDPC model.

The shear failure line of MDPC is defined in the *p–t* plane as:

$$F_s = t - p * tan\beta - d = 0 \tag{3}$$

where $d$ and $\beta$ are the cohesion and friction angle of the soil, respectively. The calibrated MDPC constitutive model parameters were derived from the consolidated undrained (CU) triaxial tests performed in a previous study [23] as shown in Table 2.

**Table 2.** The MDPC parameters of the base course layer.

| Constants | Definitions | Variable | Value |
|---|---|---|---|
| Elasticity | Elastic modulus (MPa) | $E$ | 248 |
| | Poisson's ratio | $\nu$ | 0.35 |
| Cap Plasticity | Material cohesion (kPa) | $d$ | 48.3 |
| | Angle of friction | $\beta$ | 66 |
| | Cap eccentricity | | 0.015 |
| | Initial yield surface position | | 0.005 |
| | Transition surface radius | | 0.07 |
| | Flow stress ratio | | 1 |

### 3.3. Subgrade Soil

The modified Cam–Clay (MCC) model was selected in the current study to model the behavior of natural/subgrade soil, which is considered to be soft, medium stiff, and in stiff clay conditions. The critical state concept theory serves as the foundation for the elastoplastic MCC model. The yield surface of the MCC model in the $p'$–$q$ plane is an ellipse. The magnitude of pre-consolidation pressure controls the initial size of the ellipse. The yield surface of MCC can be defined as follows:

$$f = q^2 - M_c^2 \, p \, (P_c - p) = 0 \tag{4}$$

where $q$ is the deviatoric stress, $P_c$ is the pre-consolidation pressure, and $p$ is the mean stress. The calibrated parameters for the MCC model which represent the soft, medium, and stiff clays, were determined for the existing clay soil properties in Louisiana that were derived in a previous study through experimental tests [23]. The subgrade's calibrated parameters of clay soil with various stiffness conditions for the MCC model are presented in Table 3.

**Table 3.** The MCC model's subgrade layer characteristics.

| Stiffness | G (MPa) | $\nu$ | M | $\lambda$ | $\kappa$ | $e_0$ |
|---|---|---|---|---|---|---|
| **Weak** | 7.4 | 0.4 | 0.56 | 0.173 | 0.035 | 1.5 |
| **Medium stiff** | 19.1 | 0.4 | 0.86 | 0.087 | 0.017 | 1.3 |
| **Stiff** | 31.8 | 0.4 | 1.2 | 0.043 | 0.009 | 0.7 |

For considering the loading history on clay soils in the numerical models, a decreasing over consolidation ratio (OCR) was considered for the subgrade soil, which starts from 3 at the surface and reaches 1 at a depth of 4.57 m.

### 3.4. Geosynthetics

To evaluate the effects of geosynthetic type and stiffness on the benefits of reinforcing pavement structures, both geogrid and geotextile types with three different tensile stiffnesses were used to reinforce the pavement at the interface between the base course and the subgrade layer. The difference between the geogrid and geotextile in the FE numerical models was distinguished through the simulation of the geogrid/geotextile–geomaterial interface, as will be discussed in the following section. The equivalent isotropic elastic stiffnesses, $E_{equivalent}$, of the geosynthetics were derived from the geosynthetics' orthotropic linear elastic properties according to the machine and cross-machine directions. The values of $E_{equivalent}$ for three different geosynthetic stiffness ranges (low, medium, and high) taken from Perkins et al. [26] are shown in Table 4.

**Table 4.** Geosynthetics equivalent elastic modulus for different stiffnesses.

| Geogrid and Geotextile Stiffness | v | $E_{equivalent}$ (MPa) |
|---|---|---|
| Low | 0.25 | 430.2 |
| Medium | 0.25 | 928 |
| High | 0.25 | 1259.7 |

*3.5. Interface Models between the Geotextile/Geogrid and Soil*

The friction model of Coulomb which is available in the ABAQUS software was used to simulate the interaction behavior between the geosynthetics and geomaterials. There should be no separation between the two neighboring elements. The normal interaction in Coulomb's model is reproduced with hard contact, whereas to represent the shearing behavior, a friction coefficient (μ) and elastic slip value ($E_{slip}$) were selected to simulate the tangential interaction which is simulated along with the geosynthetic-base/subgrade interface. The friction coefficient and elastic slip value for the base course and the geotextile interface were selected as 0.85 and 1 mm, respectively, while these values for the geotextile and subgrade interface were 0.75 and 1 mm, respectively. Meanwhile, the friction coefficient and elastic slip values of 1.475 and 1 mm were selected for the geogrid and base course interface, respectively, while these values for the geotextile and subgrade interface were 0.75 and 1 mm, respectively. The friction coefficients between different surfaces were selected based on the pullout tests performed in previous research [26].

The interlocked aggregates in the geogrid aperture were modeled using a newly proposed modeling technique. In this technique, the geogrid layer in this model was split into two sublayers and bonded the aggregates in between, as illustrated in Figure 3. The tensile stiffnesses of the two geogrid sublayers were taken to be half (E/2) of the geogrid stiffness. The interlocking aggregate layer was estimated to have a 10 mm thickness. This was equivalent to the crushed limestones' maximum aggregate size ($D_{max}/2$) divided by half.

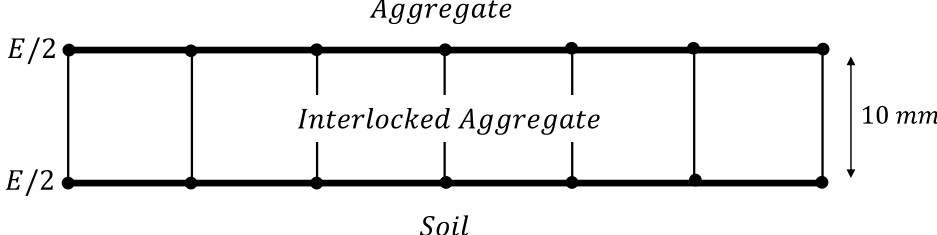

**Figure 3.** Geogrid–aggregate interlocking effect simulation.

*3.6. Confinement Effect*

In order to account for the impact of confinement on base course aggregates, Gu et al. [27] looked at several experimental findings on geosynthetic-reinforced specimens. They next put out a stress distribution plan that adequately accounts for the aggregates' increased confinement stress. This method suggests that geosynthetics under vertical loading produce an influence zone that acts as additional confinement for the aggregates, adding up to the lateral pressure brought on by compaction. For various geosynthetic types, the lateral confining stresses were estimated. This area of impact is assumed as a 7.62 cm zone located on top of the geosynthetic layer, featuring a maximum confining stress of $\Delta\sigma_{3max}$ at the level of the geosynthetic material and $\Delta\sigma_{3max}$ equal to zero at the edge. Using UMAT subroutines, at the first stage of analysis, the lateral confining pressures are taken into account in the FE models. Table 5 displays the calculated values of $\Delta\sigma_{3max}$.

**Table 5.** Maximum lateral confining stress for geosynthetics with varying levels of stiffness.

| Geogrid Stiffness | $\Delta\sigma_{3max}$ (kPa) | Geotextile Stiffness | $\Delta\sigma_{3max}$ (psi) |
|---|---|---|---|
| **Low** | 29 | Low | 25 |
| **Medium** | 31 | Medium | 29 |
| **High** | 34 | High | 32.4 |

### 3.7. Cyclic Wheel Loading

When a vehicle's speed and its kind change, the wheel loading's form, size, and duration change [28]. In this study, the MEPDG design guideline's specification NCHRP2004 was adopted for the peak pressure (80 psi) and the contact area (a circular area with a radius of 6 in) of the wheel loading in the FE numerical models [29]. A haversine-shape loading with a uniform distributed pressure was considered to simulate the wheel loading on the pavement surface, where the following equation was used to compute the load $F$ at the moment $t$:

$$F = \frac{P\left(1 - \cos\left(\frac{2\pi t}{T}\right)\right)}{2} \tag{5}$$

where 550 kPa is used as the peak pressure ($P$ = 550 kPa), the time for one loading cycle is 0.1 s ($T$ = 0.1 s), and the resting time to the next load is 0.9 s, simulating a 1 Hz frequency moving load (Figure 4).

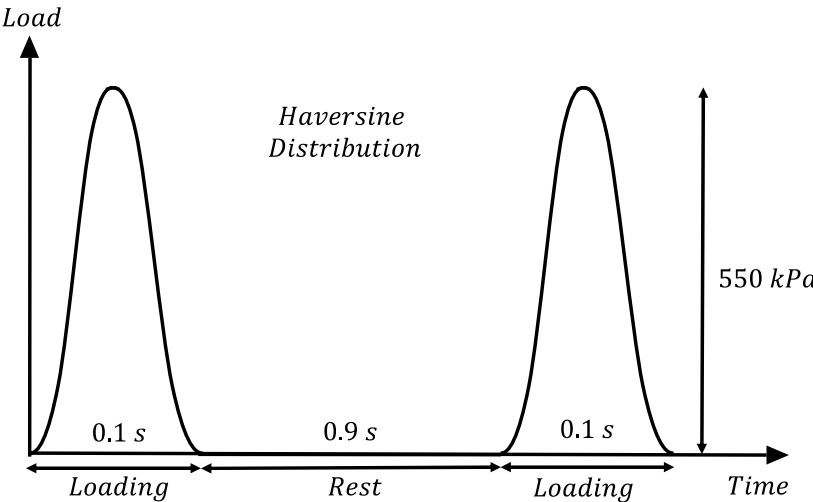

**Figure 4.** Haversine distribution of vehicular loading.

### 3.8. Finite Element Parametric Study

A total of 84 FE models were developed and evaluated during this study. The thickness of HMA was assumed to be 8.9 cm (3.5 in.) for all of the models, which equals the thickness of low-volume traffic roads in Louisiana. In order to evaluate the contribution of each variable/parameter to the pavement's responses under repeated loading, a FE parametric study was performed on different subgrade strengths/stiffnesses, different base thicknesses, two geosynthetics types, and different geosynthetics tensile moduli. As shown in Table 6, three different stiffnesses were selected for the subgrade stiffnesses, four different thicknesses were selected for the base course aggregate, and three different stiffnesses were selected for geosynthetics with two different types of geotextiles and geogrids. The location of the geosynthetic layer was assumed in this study to be at the base-subgrade interface.

**Table 6.** Parametric study.

| Subgrade Stiffness | Base Thickness (cm) | HMA Thickness (cm) | Geosynthetic Location | Geosynthetic Stiffness | Geosynthetic Type |
|---|---|---|---|---|---|
| Weak Medium stiff Stiff | 20.3 | 8.9 | Base/sub-Interface | Low Medium High | Geogrid Geotextile |
| | 25.4 | | Base/sub-Interface | | |
| | 30.5 | | Base/sub-Interface | | |
| | 35.6 | | Base/sub-Interface | | |

## 4. Methodology

As described in Figure 5, six steps were taken in the current study to calculate the structural benefits of using geosynthetic reinforcement in pavement construction. In these steps, the FE results were first calibrated and then combined with the transfer functions in the MEPDG. The benefits were evaluated using AASHTOWare software. In order to calculate the structural benefits of geosynthetics in terms of TBR, $M_{R\text{-eff}}$, and BCR, the following steps were used:

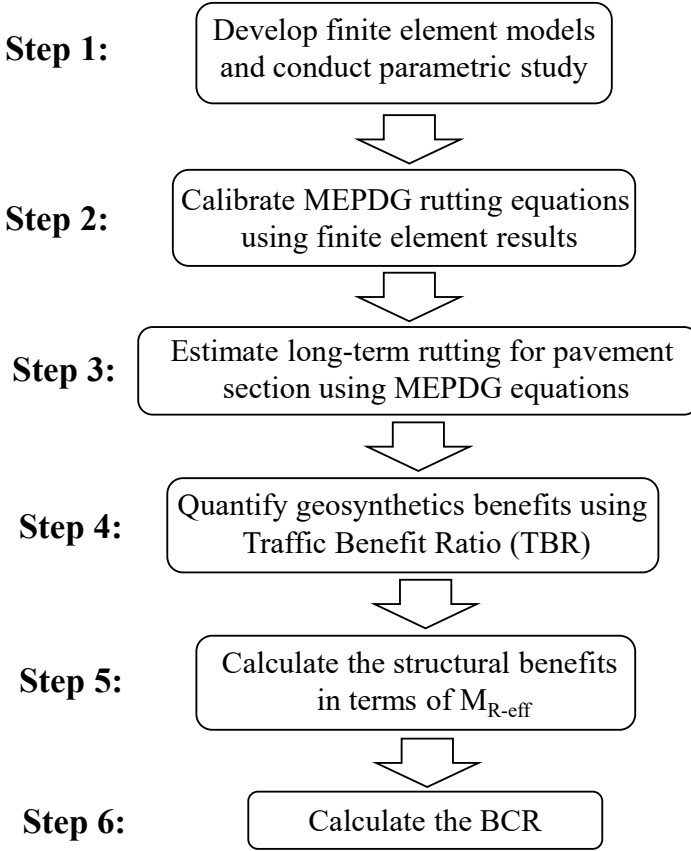

**Figure 5.** The calculation of structural benefits of the geosynthetics workflow.

In step 1, an FE parametric study was performed using the developed 2D axisymmetric quasi-static FE models. The FE models were used to simulate the geosynthetic-reinforced pavement under repeated loading for 100 cycles for low-volume roads. The developed FE models consider both the confining pressures due to the effect of geosynthetic reinforcement on the base aggregates [30]. The rutting curve during the loading cycles and the vertical resilient strain of the sublayers were obtained for each model.

In step 2, the MEPDG transfer functions (rutting equations) were then calibrated fitting the results for each sublayer rutting curve in 100 cycles using the resilient vertical strains derived in step 1. The rutting equations were fitted into rutting curves based on the criterion of the least square of error method. The MEPDG transfer functions are explained in Section 4.1.

In step 3, the calibrated rutting curves of each sublayer were extended to reach the pavement total rutting targets of 12.7, 19.05, and 25.4 mm in order to establish the rutting curves versus the loading cycles. The corresponding loading cycles for each rutting target are derived in this step.

In step 4, the traffic benefit ratio (TBR) was used to determine the long-term advantages of employing geosynthetics in pavement structures. The derivation of TBR is explained in Section 5.

In step 5, the structural benefits of reinforced sections were calculated in terms of increased effective resilient modulus of base ($M_{R\text{-eff}}$) using AASHTOWare software.

In step 6, the base course reduction (BCR) values were derived. The $M_{R\text{-eff}}$ and BCR deriving methods are explained in Section 5.

*4.1. MEPDG Transfer Functions*

The total rutting for pavement structures in the MEPDG is the sum of individual sublayers ruttings. The vertical resilient strain (elastic deformation) and the rutting for each sublayer are correlated by the transfer functions (rutting equations) in MEPDG. The transfer function for the HMA layer is given as:

$$\Delta_{HMA} = \varepsilon_{p(HMA)} h_{HMA} = \beta_{1r} k_z \varepsilon_{r(HMA)} 10^{k_{1r}} N^{k_{2r}\beta_{2r}} T^{k_{3r}\beta_{3r}} h_{HMA} \tag{6}$$

where $\Delta_{HMA}$ is the permanent rutting, $h_{HMA}$ is the thickness of the HMA layer, $\varepsilon_{p(HMA)}$ is the permanent plastic strain, $k_z$ = the factor of depth confinement, $\varepsilon_{r(HMA)}$ = the mid-depth vertical resilient strain from the model, $T$ = the temperature of the pavement, $N$ = the number of loading cycles, $k_{1r}$, $k_{2r}$, and $k_{3r}$ = the global field calibration coefficient, and $\beta_{1r}$, $\beta_{2r}$, and $\beta_{3r}$ = the local calibration coefficients.

The transfer function of the base course layer and subgrade is given in the following equation:

$$\Delta_{soil} = \beta_{s1} k_{s1} \varepsilon_v h_{soil} \left( \frac{\varepsilon_0}{\varepsilon_r} \right) e^{-\left( \frac{\rho}{N} \right)^{\beta}} \tag{7}$$

where $\Delta_{soil}$ is permanent deformation, $\beta_{s1}$ is a calibrating factor, $\varepsilon_v$ is the mid-depth and top vertical resilient strain for the base course and subgrade, $\varepsilon_0$, $\varepsilon_r$ $\beta$, and $\rho$ are material parameters, $N$ is the number of loading cycles, and $h_{soil}$ is the layer thickness of the base course or subgrade.

## 5. Calculating TBR

The use of geosynthetic reinforcement usually results in increasing the service life of pavement structures by reducing the permanent deformation due to vehicular loadings. The increased service life of pavement structures can be expressed in terms of the traffic benefit ratio (TBR), which is defined as the ratio between the required cycles of the reinforced section ($N_R$) to the unreinforced section ($N_U$) for a specific level of performance or reaching a rutting target. The rutting target is typically chosen to be 12.7, 19.05, or 25.4 mm rutting depths.

$$TBR = \frac{N_R}{N_U} \qquad (8)$$

To derive the TBR value from Equation (8), the number of cycles to reach the rutting target is derived for the pavement section without reinforcement ($N_U$) and the pavement section with reinforcement ($N_R$) from the calibrated/extended curves of numerical modeling.

*Calculation of Structural Benefits*

The derived TBRs from the previous section were incorporated into the Mechanistic-Empirical approach using the AASHTOWare 2.6.0 in order to quantify the structural benefits of geosynthetics as reinforcement in the pavement structure, in terms of effective base layer resilient modulus ($M_{R\text{-eff}}$) and bace course reduction (BCR). The following stages were adopted to quantify the structural benefits for each TBR value in the reinforced pavement section (as described in Figure 6):

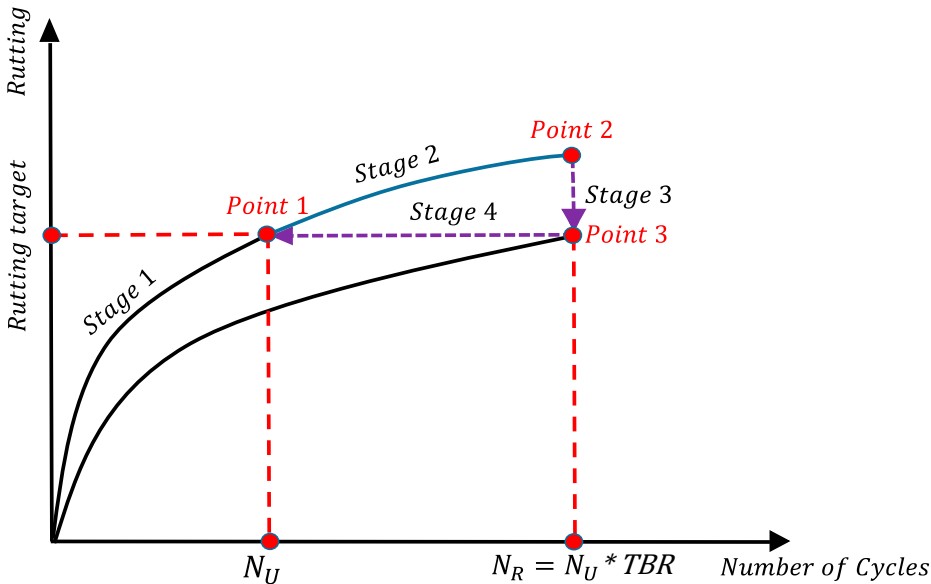

**Figure 6.** Different stages of structural benefits calculations.

In stage 1, the value of $N_U$ that corresponds to the rutting target for the unreinforced pavement section is derived. In this stage, the corresponding pavement section for a selected TBR is modeled in AASHTOWare using the Louisiana calibrated design coefficients for each sublayer. The number of average annual daily trucks (AADT) is then adjusted in the model so that the rutting curve reaches the target rutting ending in point 1. The rutting targets are equal to the rutting of selected TBRs, which can be 12.7, 19.05, or 25.4 mm.

In stage 2, the corresponding rutting value of the unreinforced section to $N_R$ value is derived. In this stage, the rutting curve at stage 1 is extended by increasing the number of load cycles from $N_U$ to $N_R$, reaching to point 2. In order to extend the curve, the derived AADT in stage 1 (AADT$_U$) is multiplied by the derived TBR from the model to derive the value of AADT$_R$ (AADT$_R$ = AADT$_U$ * TBR). The final rutting at point 2 is higher than the rutting target of the reinforced section.

In stage 3, the effective base layer resilient modulus ($M_{R\text{-eff}}$) is derived for the selected TBR. In this stage, the base layer resilient modulus ($M_R$) of the model is increased to push the rutting curve down to match the rutting target level of the reinforced section at the number of cycles equal to $N_R$. The new rutting curve in this stage is ended at point 3.

In stage 4, the base course reduction (BCR) value is derived. In this stage, the number of loadings in the model is changed to the $AADT_U$, and the base thickness with the $M_{R\text{-eff}}$ is adjusted (decreased) so that the rutting curve would be ended in point 1 (i.e., match the rut curve of the unreinforced section). The new curve endpoint is the same as the endpoint in stage1. The BCR value is the percent difference between the initial and adjusted base thickness (Equation (9)).

$$BCR = \frac{Reduction\ in\ base\ thickness}{Initial\ base\ thickness} * 100 \tag{9}$$

## 6. Results and Discussion

### 6.1. Permanent Deformations and Verification

As discussed in Section 3.1, the MEPDG provides transfer functions for each sublayer of the pavement structure. The rutting curves (permanent deformation) of each sublayer from different FE models for the first 100 load cycles are used to calibrate the MEPDG transfer functions (rutting equations). Each sublayer's calibrated transfer function was then used to derive the long-term rutting curve. The summation of derived long-term ruttings of all of the sublayers will produce the long-term total surface rutting of the pavement structure. In accordance with the calibrated rutting equations, Figure 7 illustrates instances of the M-E calibrated rutting curves from the first 100 loading cycles of the FE models and the long-term extrapolated M-E predictions for the unreinforced, geogrid-reinforced, and geotextile-reinforced sections for the base thicknesses of 30.5 cm and geosynthetics of medium stiffness.

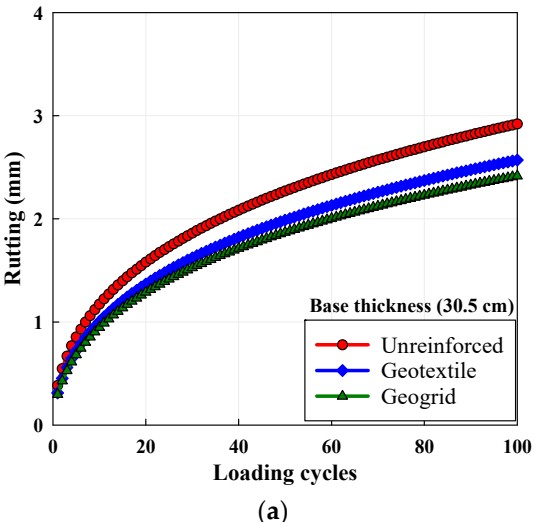
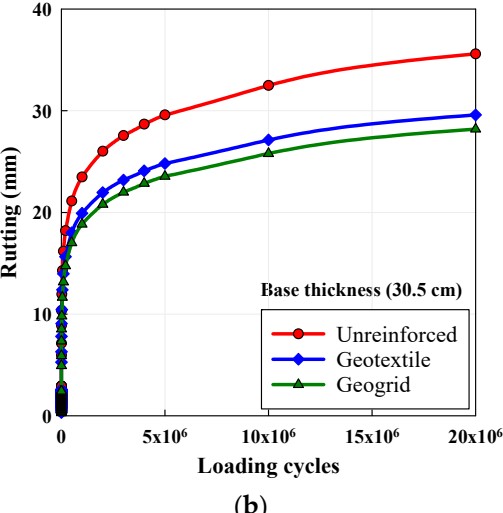

(**a**)  (**b**)

**Figure 7.** The example of (**a**) calibrated ruttings for 100 cycles and (**b**) extrapolated ruttings.

The full-scale findings in a prior investigation at the Pavement Research Facility (PRF) site on geosynthetic-reinforced test lane sections using cyclic plate load testing (CPLT) were used to verify the M-E extrapolated long-term rutting curves [13]. At the PRF site, on weak subgrade soil (CBR = 1.5), six test lane sections with a length of 24.4 m and width of 4 m were constructed. The extrapolated calibrated FE results were verified by the lane's sections number 3,4 and 6 from the PRF. The long-term rutting curves from the FE extrapolation were compared with the measured rutting curves from the CPLTs, as shown in Figure 8. The results demonstrated good agreement between the FE extrapolated rutting curves and the CPLT rutting curves.

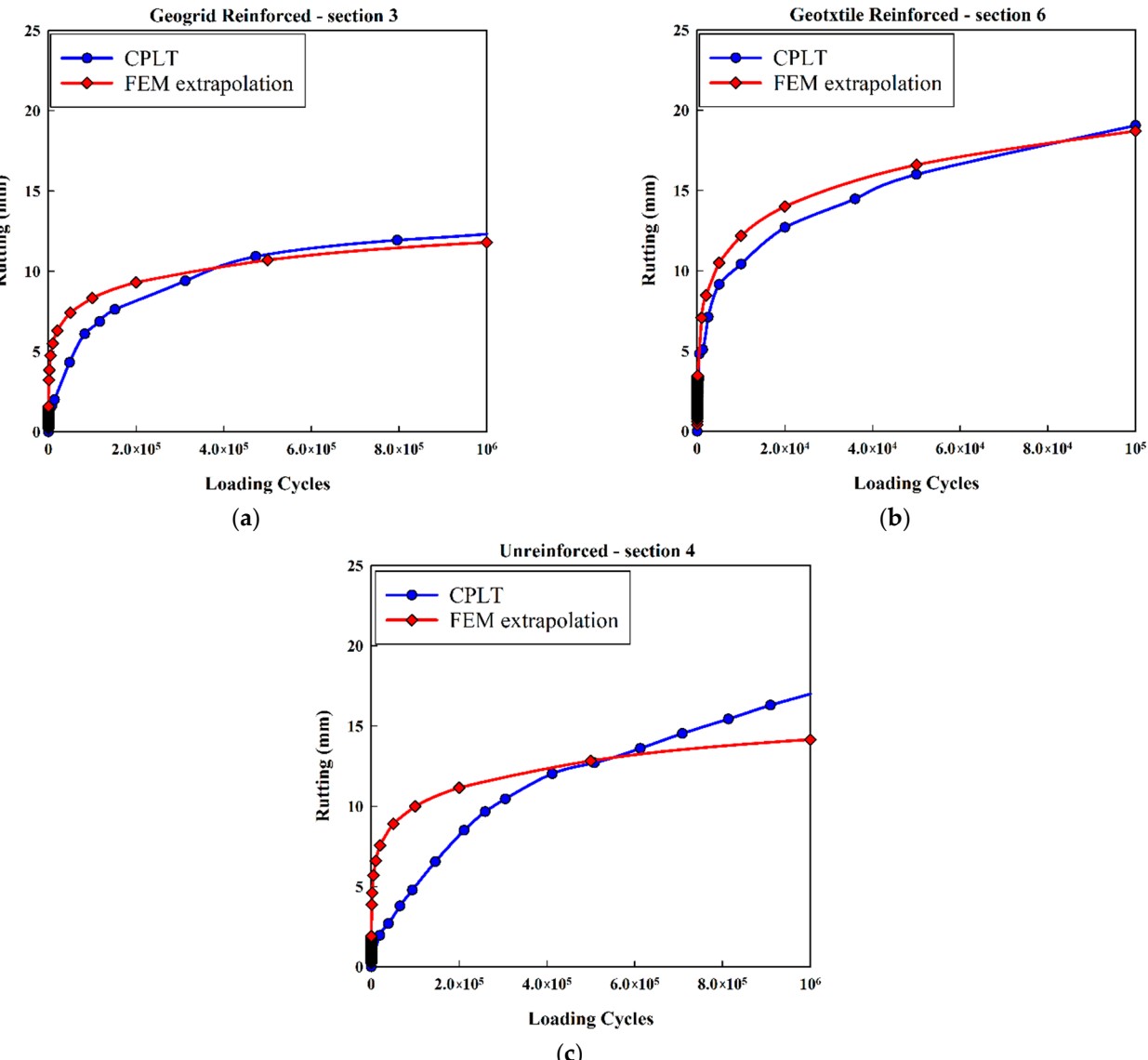

**Figure 8.** The comparison of the FE extrapolated rutting curves and those of CPLT for (**a**) a 49.28 cm base thickness with a geogrid, (**b**) a 26.67 cm base thickness with a geotextile, and (**c**) a 49 cm base thickness without reinforcement.

### 6.2. Traffic Benefit Ratios (TBRs)

One approach to quantify the benefits of using geosynthetic reinforcement in pavement structures is by evaluating the TBR. The TBR value represents the increased ratio of the pavement's service life due to the inclusion of geosynthetics for a selected level of performance or a rutting target. For three different levels of performance (corresponding to rutting depths of 12.7, 19.05, or 25.4 mm), the TBR values were derived from the FE extended rutting curves for the different reinforced sections using the procedure explained in Section 4. Figures 9–11 present the variations of derived TBR values versus the base course thickness for different rutting depths, geosynthetics types, geosynthetics tensile stiffnesses, and subgrade strengths/stiffnesses. The figures clearly demonstrated that the TBR values increase with the increasing rutting target. The differences in TBRs for different rutting targets are usually higher for geogrid-reinforced cases as compared to geotextile-reinforced cases, with a maximum difference of up to 96% at a 25.4 cm base thickness on a weak subgrade soil. The difference between the TBR for the geogrid-reinforced sections at 25.4 cm base thickness for the medium stiff and stiff subgrades is 73% and 37%, respectively.

The thickness of the base course layer also shows a significant effect on the TBR values. The TBR values for all of the cases reach a maximum (optimal) value at a 25.4 cm base thickness. Reaching a peak value for TBR due to changing the base thickness was also reported by other researchers [31,32]. Apparently, the benefits of using geosynthetics to reinforce pavements show an optimal TBR value for a base course thickness of 25.4 cm.

For all cases, the type of geosynthetic material (geogrids versus geotextiles) has an important effect on the TBR benefits, and the differences between TBR values are the highest for the weak subgrade as compared to the other subgrade stiffnesses. The comparison between the calculated TBR values for various reinforced pavement sections shows that geogrids have higher TBR values than geotextiles (up to 40%). The higher TBR values for geogrids can be linked to the behavior of interlocking (that was modeled in this study) between the geogrid apertures and the base aggregate, which resulted in consistent rutting behavior with the accelerated field test sections [13].

The effect of the geosynthetics' tensile stiffness was also evaluated for all of the geosynthetic-reinforced cases. For the geogrid-reinforced cases, the TBR values for the low stiffness cases range from 1.35 to 5.3, and for the high stiffness cases, they range from 1.74 to 8.9. The changes in geogrid stiffness from low to high for the geogrid-reinforced cases result in an increase in TBR value of up to 68%. However, for the geotextile-reinforced cases, the TBR values for the low-stiffness cases range from 1.25 to 3.75, and for the high-stiffness cases, they range from 1.58 to 5.92. The changes in geotextile stiffness from low to high for the geotextile-reinforced cases result in up to a 54 % increase in TBR value.

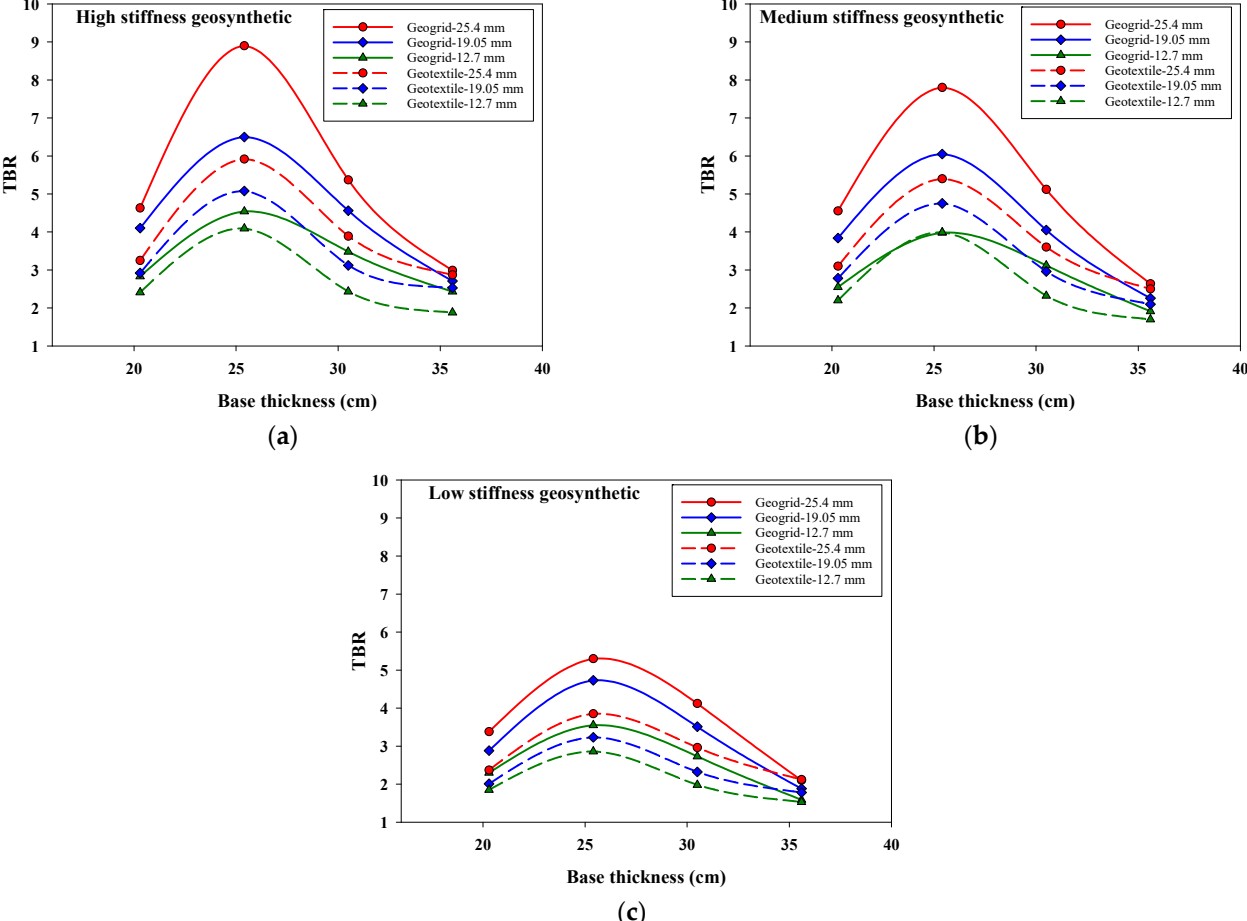

**Figure 9.** TBR variations with base thickness for pavements reinforced with a single layer of (**a**) high, (**b**) medium, and (**c**) low stiffness geosynthetics on the weak subgrade.

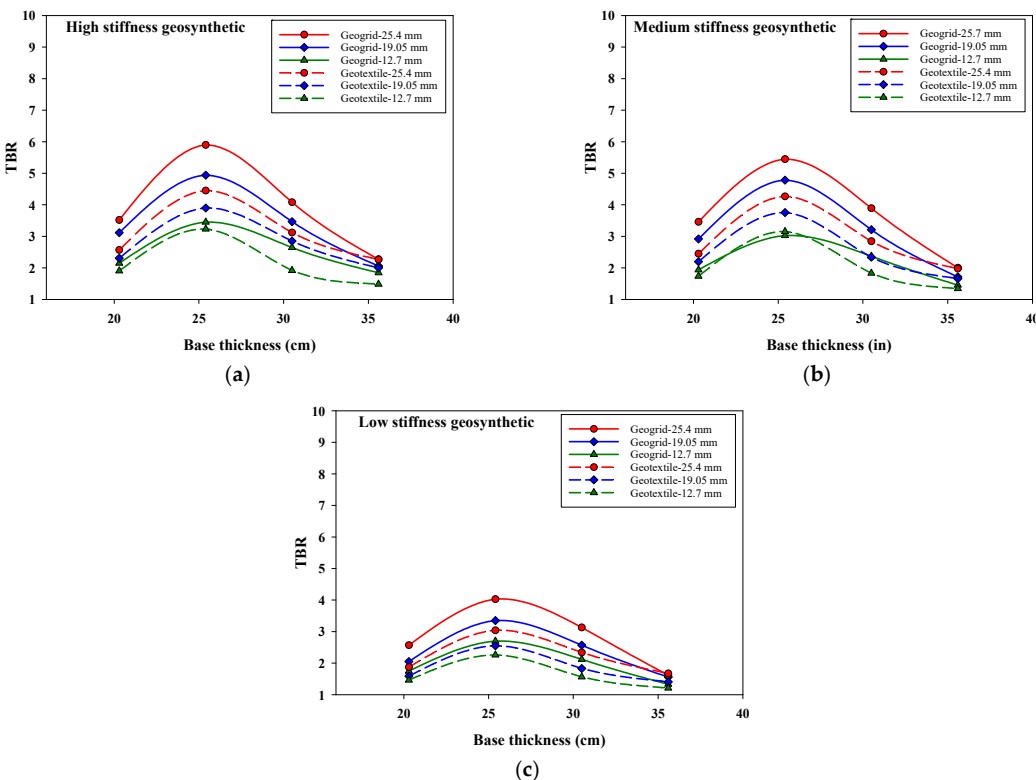

**Figure 10.** TBR variations with base thickness for pavements reinforced with a single layer of (**a**) high, (**b**) medium, and (**c**) low stiffness geosynthetics on the medium stiff subgrade.

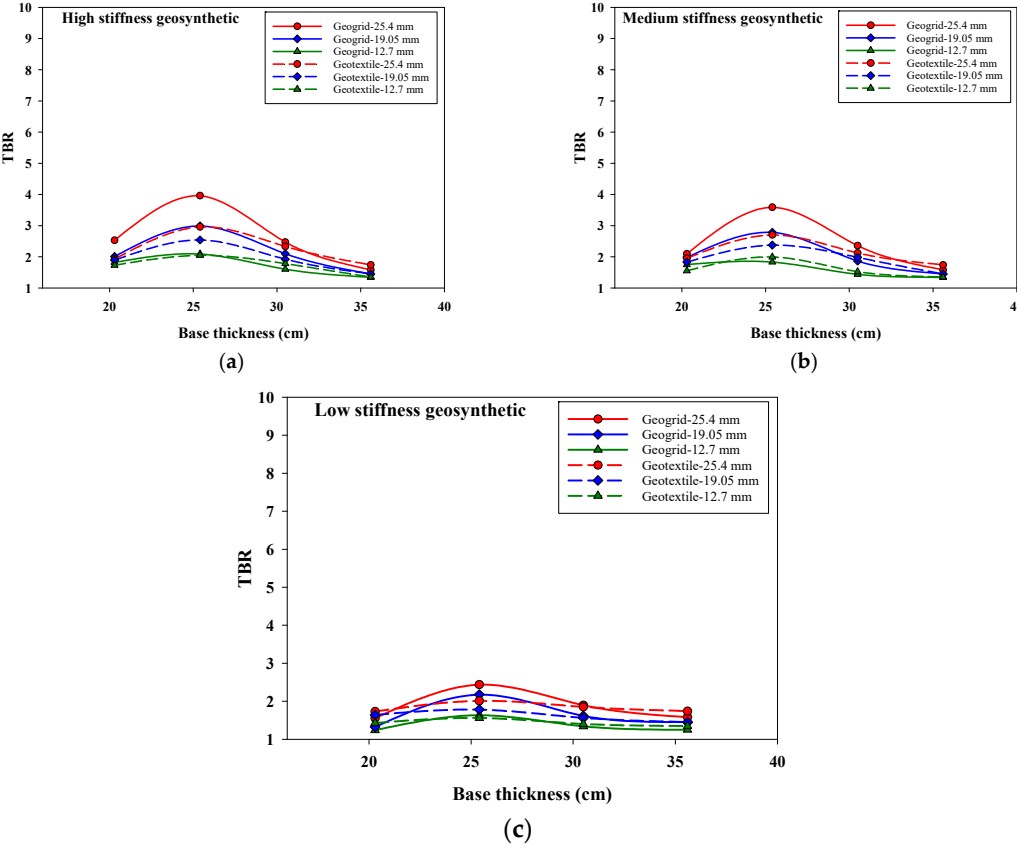

**Figure 11.** TBR variations with base thickness for pavements reinforced with a single layer of (**a**) high, (**b**) medium, and (**c**) low stiffness geosynthetics on the stiff subgrade.

*6.3. Effective Base Resilient Modulus ($M_{R\text{-}eff}$)*

One approach to incorporate the benefits of geosynthetic reinforcement into the ME design of flexible pavements is to assume that all the geosynthetic benefits (for design purposes) go into reinforcing the base course layer only. In this method, the resilient modulus of the whole base layer thickness will be increased from $M_R$ to $M_{R\text{-}eff}$, as explained in Section 4.1 (stage 3), such that the rutting curve of the unreinforced section (with $M_{R\text{-}eff}$) is pushed down to match the rutting target of the geosynthetic-reinforced section. In this approach, the properties of other sublayers will be the same as in the corresponding section.

As explained in Section 3.3, the calculated TBR values from FE analysis are used as an input in AASHTOWare to derive the corresponding $M_{R\text{-}eff}$ values for the base layer. As a result, the variation in $M_{R\text{-}eff}$ values follows the same pattern as the TBR values that shows a peak at 25.4 cm base thickness. The derived $M_{R\text{-}eff}$ values are also higher for higher rutting targets. The results of the AASHTOWare analysis show that the effective base modulus ($M_{R\text{-}eff}$) can be increased up to 322% for the geosynthetic-reinforced sections. In addition, the results show that the value of $M_{R\text{-}eff}$ increases with increasing the subgrade strength/stiffness, it increases with increasing the geosynthetic tensile modulus, and it increases with decreasing the based thickness. For all of the cases, $M_{R\text{-}eff}$ is higher for geogrid reinforcement than geotextile reinforcement. Table 7 summarizes the results of the $M_{R\text{-}eff}$ values for all of the cases.

**Table 7.** Derived $M_{R\text{-}eff}$ increased values for reinforced cases over a weak subgrade soil.

| Type | Gesy. Stiffness | Base Thickness (cm) | $M_{R\text{-}eff}$ Increase (%) | | | | | | | | |
| --- | --- | --- | --- | --- | --- | --- | --- | --- | --- | --- | --- |
| | | | Weak Subgrade | | | Medium Stiff Subgrade | | | Stiff Subgrade | | |
| | | | 12.7 mm | 19.05 mm | 25.4 mm | 12.7 mm | 19.05 mm | 25.4 mm | 12.7 mm | 19.05 mm | 25.4 mm |
| Geotextile | High tensile stiffness | 20.3 | 77 | 122 | 150 | 56 | 81 | 94 | 25 | 33 | 42 |
| | | 25.4 | 136 | 222 | 283 | 103 | 131 | 153 | 33 | 58 | 81 |
| | | 30.5 | 100 | 133 | 175 | 42 | 83 | 92 | 19 | 25 | 44 |
| | | 35.6 | 44 | 67 | 83 | 14 | 56 | 67 | 11 | 17 | 33 |
| | Medium tensile stiffness | 20.3 | 67 | 83 | 94 | 50 | 62 | 89 | 17 | 31 | 39 |
| | | 25.4 | 136 | 169 | 186 | 94 | 129 | 142 | 25 | 50 | 69 |
| | | 30.5 | 61 | 89 | 117 | 36 | 58 | 75 | 11 | 33 | 44 |
| | | 35.6 | 31 | 50 | 61 | 11 | 33 | 53 | 8 | 14 | 19 |
| | Low tensile stiffness | 20.3 | 44 | 50 | 67 | 22 | 28 | 50 | 3 | 14 | 17 |
| | | 25.4 | 89 | 106 | 133 | 56 | 64 | 89 | 14 | 19 | 25 |
| | | 30.5 | 44 | 61 | 89 | 17 | 25 | 50 | 8 | 14 | 25 |
| | | 35.6 | 17 | 33 | 50 | 8 | 17 | 31 | 6 | 14 | 19 |
| Geogrid | High tensile stiffness | 20.3 | 94 | 136 | 155 | 81 | 144 | 164 | 25 | 44 | 69 |
| | | 25.4 | 156 | 239 | 322 | 161 | 222 | 277 | 36 | 67 | 100 |
| | | 30.5 | 111 | 156 | 186 | 108 | 152 | 180 | 14 | 47 | 67 |
| | | 35.6 | 67 | 77 | 100 | 58 | 78 | 97 | 8 | 17 | 22 |
| | Medium tensile stiffness | 20.3 | 77 | 122 | 150 | 81 | 133 | 156 | 22 | 42 | 47 |
| | | 25.4 | 136 | 222 | 283 | 136 | 211 | 261 | 31 | 61 | 83 |
| | | 30.5 | 100 | 133 | 175 | 89 | 136 | 172 | 11 | 39 | 61 |
| | | 35.6 | 44 | 67 | 83 | 31 | 67 | 86 | 8 | 14 | 19 |
| | Low tensile stiffness | 20.3 | 17 | 89 | 105 | 50 | 64 | 89 | 3 | 22 | 36 |
| | | 25.4 | 105 | 156 | 178 | 83 | 111 | 142 | 19 | 47 | 58 |
| | | 30.5 | 72 | 111 | 133 | 44 | 67 | 94 | 8 | 28 | 27 |
| | | 35.6 | 22 | 42 | 61 | 11 | 28 | 31 | 6 | 14 | 22 |

6.3.1. Effect of Subgrade Stiffness on $M_{R\text{-eff}}$

The derived $M_{R\text{-eff}}$ values show that the subgrade strength/stiffness greatly impacts the results, as reflected by the TBR values. For the geotextile-reinforced sections and the 19.05 mm rutting target, the maximum increase in $M_{R\text{-eff}}$ (corresponding to the base thickness of 25.4 cm) for the high tensile stiffness geotextile ranges from 222% to 81% as the subgrade stiffness changes from weak to stiff (Figure 12). However, the values of maximum $M_{R\text{-eff}}$ change from 169% to 69% for the medium tensile stiffness geotextile and from 106% to 25% for low tensile stiffness cases (see Table 6). For the geogrid-reinforced sections, the results show a similar pattern. For high tensile geogrid and 19.05 mm rutting target, the maximum increase in $M_{R\text{-eff}}$ is 239% for the weak subgrade and 67% for the stiff subgrade. However, for the medium and low tensile stiffness and 19.05 mm rutting target, the maximum increase in $M_{R\text{-eff}}$ values changes from 222% to 156% and from 61% to 47%, respectively.

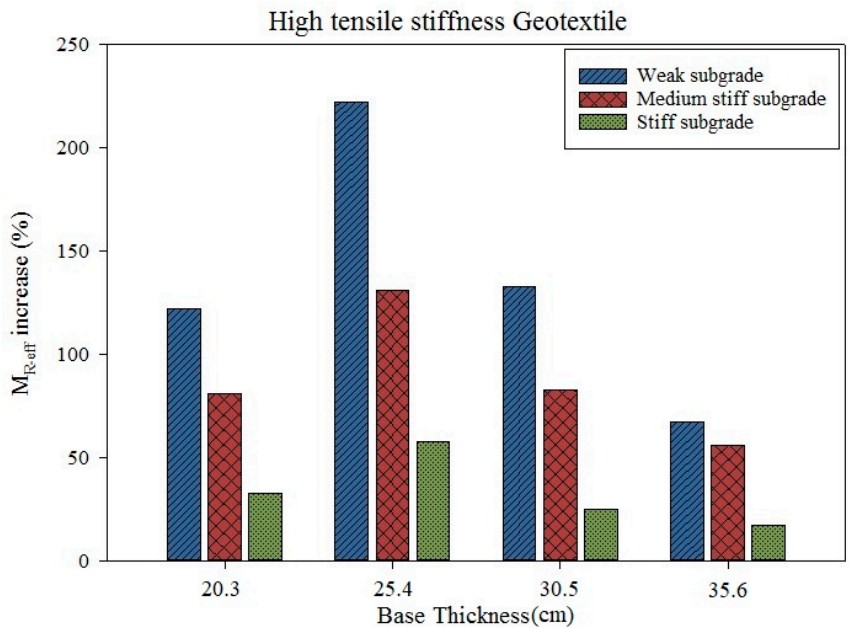

**Figure 12.** Changes in $M_{R\text{-eff}}$ increase of pavement reinforced with a high tensile stiffness geotextile for different subgrade stiffnesses.

6.3.2. Effect of Geosynthetic Type on $M_{R\text{-eff}}$

The comparison between the geogrid- and geotextile-reinforced cases shows that the sections reinforced with geogrids always have higher $M_{R\text{-eff}}$ values than those reinforced with geotextiles. The maximum $M_{R\text{-eff}}$ value at the 19.05 mm rutting target for pavements on weak subgrade soil using high tensile stiffness geosynthetics increases from 222% to 239% by changing the geosynthetic type from a geotextile to a geogrid (Figure 13). Meanwhile, for the same pavement section with medium and low tensile stiffness geosynthetics, the maximum value of $M_{R\text{-eff}}$ increases from 169% to 222% and from 106% to 156% by changing the geosynthetic type from a geotextile to a geogrid, respectively.

For the pavement sections on the medium stiff subgrade soil and the 19.05 mm rutting target (see Table 6), the maximum value of $M_{R\text{-eff}}$ also increases by changing the geosynthetic type from a geotextile to a geogrid. These changes range from 131% to 222% for high tensile stiffness geosynthetics, from 129% to 211% for medium tensile stiffness geosynthetics, and from 64% to 111% for low stiffness geosynthetics.

The change in the $M_{R\text{-eff}}$ value due to the change in the geosynthetic type for pavements on stiff subgrade soil also follows a similar trend. The maximum increase in $M_{R\text{-eff}}$ values by changing the geosynthetic type from a geotextile to a geogrid ranges from 58% to 67% for high tensile stiffness geosynthetics, from 50% to 61% for medium tensile stiffness geosynthetics, and from 19% to 47% for low stiffness geosynthetics.

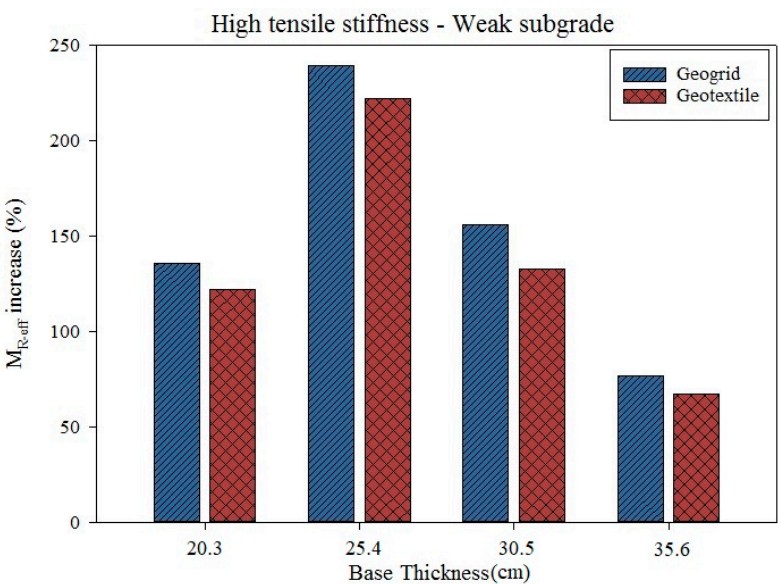

**Figure 13.** Changes in $M_{R\text{-eff}}$ increase of pavement reinforced with high tensile stiffness geosynthetics on weak subgrade soils for different geosynthetic types.

6.3.3. Effect of Geosynthetics Stiffness on $M_{R\text{-eff}}$

The effect of the geosynthetic tensile modulus on $M_{R\text{-eff}}$ is evaluated in this section. For the geotextile-reinforced cases, the change in the geotextile tensile modulus from low to high at 19.05 mm rutting for pavements on weak subgrade soil increases the maximum $M_{R\text{-eff}}$ value from 106% to 222%. For the same pavement sections on the medium stiff and stiff subgrades, the maximum value of $M_{R\text{-eff}}$ increases from 64% to 131% and from 19% to 58% by changing the geotextile tensile modulus from low to high, respectively (see Figure 14). However, for the geogrid-reinforced cases, the change in the geogrid tensile modulus from low to high at 19.05 mm rutting for pavements on weak subgrade soil results in increasing the maximum $M_{R\text{-eff}}$ value from 156% to 239% (see Table 6). For the same pavement sections on medium stiff and stiff subgrade soils, the value of $M_{R\text{-eff}}$ increases from 111% to 222% and from 47% to 67% by changing the geogrid tensile modulus from low to high, respectively.

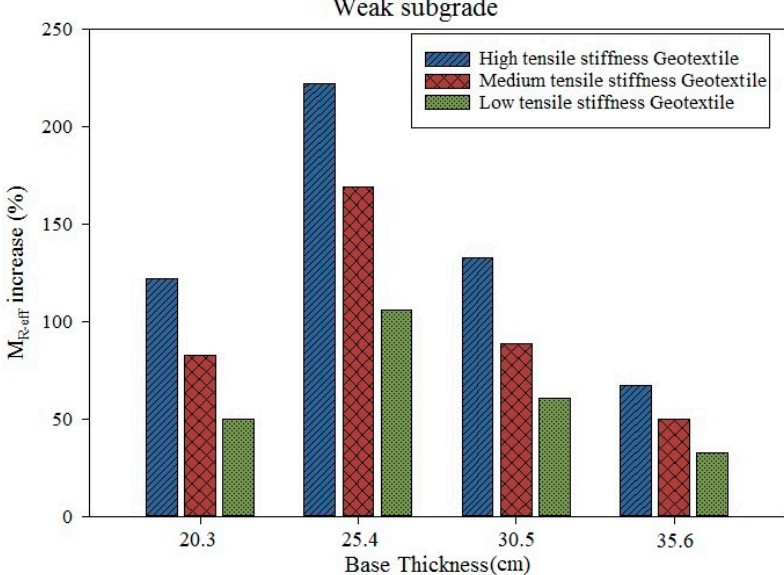

**Figure 14.** Changes in $M_{R\text{-eff}}$ increase of pavement reinforced with a geotextile on weak subgrade soil for different geosynthetic stiffnesses.

### 6.4. Base Course Reduction (BCR)

Another approach to quantify the structural benefits of geosynthetic reinforcement in flexible pavements is by evaluating the reduction in the base course layer thickness as compared to the corresponding thickness of the unreinforced section. In this method, the number of loadings (in terms of EASLs) of the reinforced section (service life) with a resilient modulus equal to $M_{R\text{-eff}}$ is assumed to be equal to the corresponding unreinforced section, as explained in Section 4.1 (stage 4). The structural benefits of geosynthetics are presented in terms of base course reduction (BCR), which is defined here as the percent reduction (or saving) in base thickness for the reinforced section of the same service life as the unreinforced section (Table 8).

**Table 8.** Derived BCR values for reinforced cases over weak subgrade soils.

| Type | Gesy. Stiffness | Base Thickness (cm) | BCR (%) | | | | | | | | |
|------|------|------|------|------|------|------|------|------|------|------|------|
| | | | Weak Subgrade | | | Medium Stiff Subgrade | | | Stiff Subgrade | | |
| | | | 12.7 mm | 19.05 mm | 25.4 mm | 12.7 mm | 19.05 mm | 25.4 mm | 12.7 mm | 19.05 mm | 25.4 mm |
| Geotextile | High tensile stiffness | 20.3 | 28 | 34 | 38 | 22 | 28 | 33 | 17 | 22 | 25 |
| | | 25.4 | 42 | 47 | 53 | 34 | 39 | 41 | 29 | 34 | 37 |
| | | 30.5 | 32 | 36 | 40 | 25 | 30 | 31 | 17 | 22 | 27 |
| | | 35.6 | 26 | 34 | 39 | 25 | 29 | 31 | 17 | 20 | 24 |
| | Medium tensile stiffness | 20.3 | 30 | 31 | 36 | 21 | 26 | 31 | 14 | 20 | 24 |
| | | 25.4 | 39 | 45 | 50 | 32 | 36 | 39 | 29 | 33 | 36 |
| | | 30.5 | 27 | 33 | 37 | 23 | 26 | 30 | 12 | 17 | 18 |
| | | 35.6 | 21 | 24 | 32 | 24 | 27 | 29 | 13 | 15 | 19 |
| | Low tensile stiffness | 20.3 | 21 | 25 | 30 | 16 | 20 | 26 | 12 | 18 | 23 |
| | | 25.4 | 32 | 35 | 39 | 27 | 30 | 35 | 13 | 15 | 19 |
| | | 30.5 | 23 | 25 | 28 | 19 | 21 | 26 | 9 | 12 | 16 |
| | | 35.6 | 16 | 20 | 23 | 19 | 22 | 26 | 10 | 14 | 16 |
| Geogrid | High tensile stiffness | 20.3 | 31 | 41 | 49 | 23 | 31 | 37 | 24 | 29 | 35 |
| | | 25.4 | 43 | 52 | 59 | 39 | 45 | 49 | 35 | 40 | 47 |
| | | 30.5 | 35 | 45 | 53 | 29 | 37 | 43 | 26 | 29 | 34 |
| | | 35.6 | 26 | 37 | 40 | 23 | 33 | 38 | 26 | 29 | 31 |
| | Medium tensile stiffness | 20.3 | 29 | 39 | 45 | 22 | 28 | 35 | 23 | 27 | 33 |
| | | 25.4 | 41 | 50 | 54 | 37 | 43 | 46 | 33 | 38 | 45 |
| | | 30.5 | 33 | 41 | 49 | 27 | 35 | 41 | 23 | 27 | 32 |
| | | 35.6 | 24 | 31 | 36 | 22 | 31 | 37 | 26 | 28 | 30 |
| | Low tensile stiffness | 20.3 | 23 | 33 | 39 | 18 | 23 | 28 | 12 | 18 | 22 |
| | | 25.4 | 36 | 42 | 49 | 30 | 34 | 38 | 28 | 33 | 37 |
| | | 30.5 | 28 | 37 | 44 | 22 | 26 | 30 | 15 | 21 | 26 |
| | | 35.6 | 19 | 28 | 31 | 22 | 26 | 29 | 13 | 16 | 21 |

### 6.4.1. Effect of Subgrade Stiffness on BCR

The subgrade stiffness has an important effect on the derived BCR values. For geotextile reinforcement, the maximum BCR (corresponding to the base thickness of 25.4 cm) for high tensile stiffness cases at 19.05 mm rutting varies from 47% for weak subgrade soil to 34% for stiff subgrade soil (Figure 15). These values change from 45% to 33% for the medium tensile stiffness geotextile and from 35% to 15% for the low tensile stiffness geotextile (see Table 6). For the geogrid reinforcement, the results show a similar trend in BCR. For the high tensile geogrid, the maximum increase in BCR at 19.05 mm rutting is 52% for the weak subgrade, while this value decreases to 40% for the stiff subgrade. For the medium and low tensile stiffness, the BCR values at 0.75 in rutting change from 50% and 38% to 42% and 33% for weak and stiff subgrades, respectively.

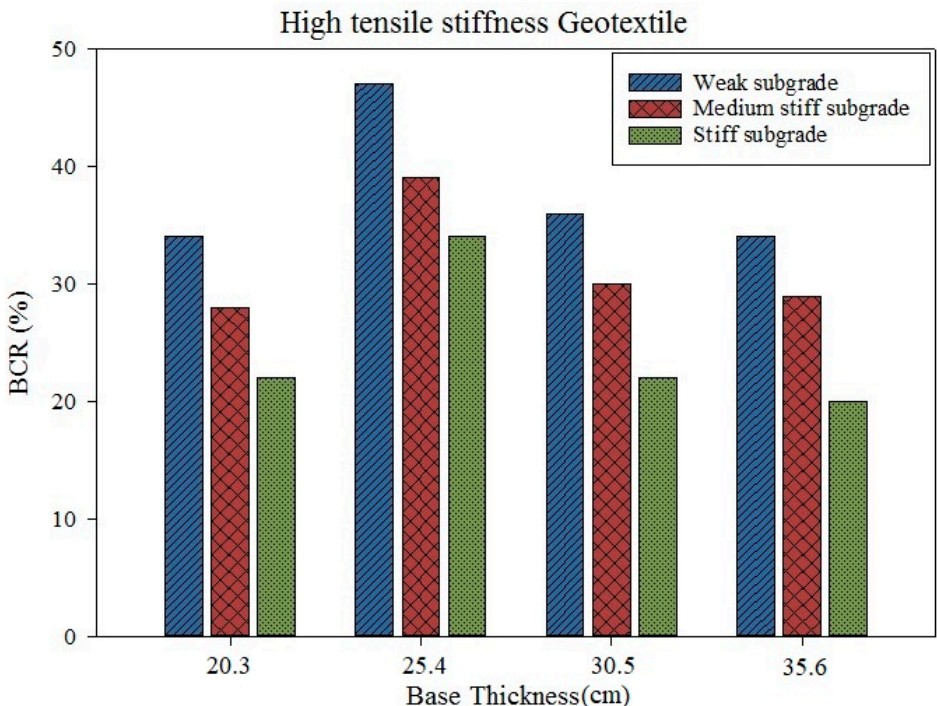

**Figure 15.** Changes in the BCR of the pavement reinforced with a high tensile stiffness geotextile for different subgrade stiffnesses.

### 6.4.2. Effect of Geosynthetic Type on BCR

The comparison of BCR values between the geogrid and geotextile reinforcements shows that the geogrids have higher BCR values than the geotextiles. The maximum value of BCR at 19.05 mm rutting for pavements on weak subgrade soil for the high tensile stiffness geotextile is 47%; the BCR value is 52% for the high tensile stiffness geogrid (Figure 16). For the same pavement sections but using medium or low tensile stiffness geosynthetics, the maximum BCR value at 19.05 mm rutting is 45% and 35% for geotextiles, while these values are 50% and 42% for the geogrids.

For the cases of pavements on the medium stiff subgrade soil and 19.05 mm rutting, the BCR values increase from 39% to 45% for high geosynthetic stiffness, from 36% to 43% for medium geosynthetic stiffness, and from 30% to 34% for low geosynthetic stiffness by changing the geosynthetic type from a geotextile to a geogrid (see Table 7). For the same sections and conditions on the stiff subgrade, the BCR values increase from 34% to 40% for high tensile stiffness, from 33% to 38% for medium tensile stiffness, and from 15% to 33% for low stiffness geosynthetics.

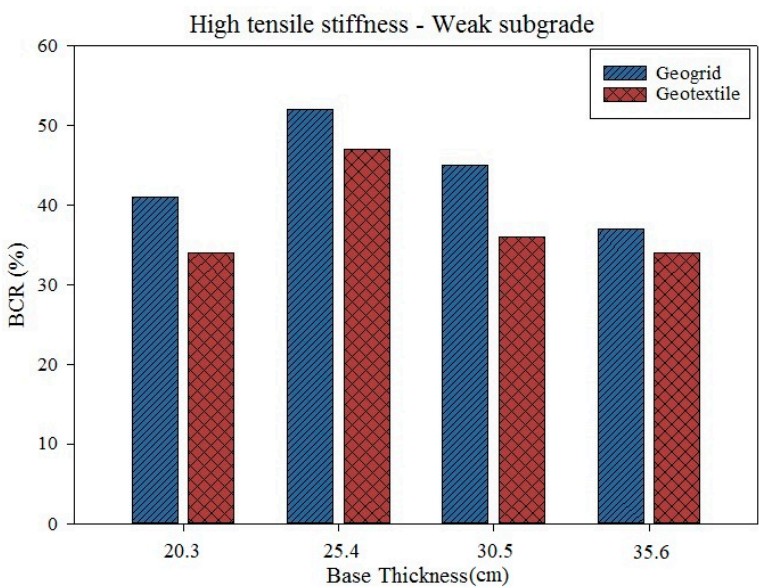

**Figure 16.** Changes in the BCR of the pavement reinforced with a high tensile stiffness geosynthetic on weak subgrade soil for different geosynthetic types.

### 6.4.3. Effect of Geosynthetics Stiffness on BCR

The results of this study demonstrate the effect of the geosynthetic tensile modulus on the BCR value. For the geotextile-reinforced cases, the increase in the geotextile tensile modulus from low to high for pavements on weak subgrade soil results in increasing the maximum BCR value from 45% to 47%. Meanwhile, the derived maximum BCR values for the pavement sections on medium and stiff subgrades increase from 30% to 39% and from 15% to 34% by changing the geotextile tensile modulus from low to high, respectively (Figure 17). Similarly, changing the geogrid tensile modulus from low to high for the geogrid-reinforced pavements on weak subgrade soil results in increasing the maximum BCR value at 19.05 mm rutting from 42% to 52% (see Table 7). For the same pavement sections on medium and stiff subgrades, the values of BCR at 19.05 mm rutting change from 34% to 45% and from 33% to 40% by changing the geogrid tensile modulus from low to high, respectively.

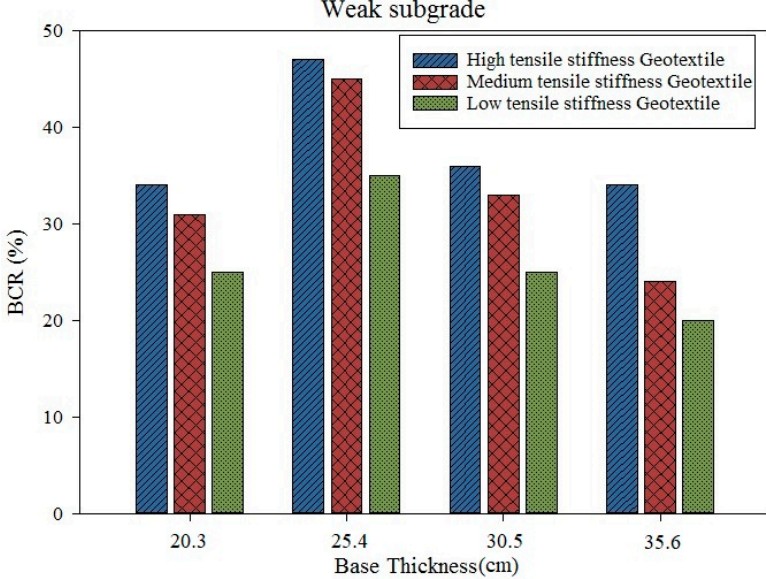

**Figure 17.** Changes in the BCR of the pavement reinforced with a geotextile on weak subgrade soil for different geosynthetic stiffnesses.

### 7. Summary and Conclusions

Several FE models have been developed using ABAQUS software to assess the effect of various parameters such as geosynthetic type and tensile stiffness, subgrade stiffness, and base thickness on the structural advantages of flexible pavements with geosynthetic reinforcement for low-traffic roads (HMA thickness of 8.9 cm). The rutting curve for the first 100 cycles was calibrated using the M-E transfer functions and then used to extrapolate the long-term rutting curves. The advantages of geosynthetic reinforcement were first calculated as traffic benefit ratio values (TBR). The structural benefits as effective base layer resilient modulus ($M_{R-eff}$) and base course reduction (BCR) factor for the base course layer were then calculated using the AASHTOWare. The calculated structural benefits hold great promise and potential for new solutions and addressing critical challenges in geosynthetic reinforced pavement (GRP) design. Despite the fact that these benefits have yet to be put into practice, they show a promising outlook for the future. The potential impact of these advancements cannot be understated, as they have the ability to bring about substantial improvement and progress in the field. It is important to keep in mind that the findings of the current research are limited to low-volume roads with an asphalt layer thickness of 8.9 cm. The full extent of the geosynthetic's structural benefits will be explored in the future by conducting studies on medium- and high-volume roads with thicker asphalt layers.

The findings of this study resulted in the following conclusions:

The addition of one layer of geosynthetic material at the base/subgrade interface can significantly enhance the rutting performance of pavement structures. The calculated benefits can be up to 8.9 in terms of TBR, 322% in terms of $M_{R-eff}$, or 64% in terms of BCR.

Geogrid reinforcement usually results in higher benefits than geotextile reinforcement. The advantage of a geogrid over a geotextile can be explained by the interlocking effect of the geogrid and base aggregates. For geogrids, the TBR values are up to 40%, the $M_{R-eff}$ values are up to 40%, and the BCR is up to 35% higher than those for geotextiles.

The selected performance level in terms of rutting targets highly affects the structural benefits. The TBR, $M_{R-eff}$, and BCR values would increase with the increase in the rutting target.

The variations of TBR, $M_{R-eff}$, and BCR values with changing base thickness demonstrate peak values at the 10 in. base thickness of pavement structures for low-volume roads.

The results show that the structural benefits increase with increasing the geosynthetic tensile stiffness. The variation in geosynthetics' tensile stiffness from low to high stiffness would increase $M_{R-eff}$ values up to 66% and the BCR values up to 114%.

**Author Contributions:** Conceptualization, methodology, validation, formal analysis, data duration and interpretation of results: M.Z. and M.A.-F.; Writing—original draft preparation: M.Z.; Writing—review & editing and supervision: M.A.-F. and G.Z.V. All authors have read and agreed to the published version of the manuscript.

**Funding:** This research study was funded by Louisiana Department of Transportation and Development and the Louisiana Transportation Research Center (LTRC Project No. 20-3GT), grant number DOTLT1000346. The APC was funded by the Louisiana State University Library Open Access Author fund.

**Data Availability Statement:** The data supporting the findings of this study are available from the corresponding author upon reasonable request.

**Acknowledgments:** The Louisiana Department of Transportation and Development and the Louisiana Transportation Research Center (LTRC Project No. 20-3GT) are the sponsors of this study (State Project No. DOTLT1000346). Zhongjie Zhang and LA DOTD engineers provided significant assistance and support to the authors in conducting this work, for which they are grateful.

**Conflicts of Interest:** The authors declare no conflict of interest.

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
