# Peer review of "Incorporating the Benefits of Geosynthetic into MEPDG"

_infrastructures, doi:10.3390/infrastructures8020035_

Round 1

Reviewer 1 Report

Section 2.6 Confinement Effect is poorly explained. Please provide more detail.

The results are limited to a pavement structure with 8.9cm of HMA. This is a rather significant limitation for most State DOT roads. This limitation should be made very clear in the paper.

The predicted rutting curves from two model runs are compared to test sections constructed at LTRC. These test sections contained a geogrid and a geotextile. A comparison of a predicted rutting curve to an unreinforced test section was not provided, but should be. This is important since the focus of the paper is on reinforcement benefit, which is only known by comparison of reinforced sections to unreinforced sections. 

Author Response

Ref.: Paper, infrastructures-2145436

INCORPORATING THE BENEFITS OF GEOSYNTHETIC INTO MEPDG

Murad Abu-Farsakh, Mehdi Zadehmohamad, and George Voyiadjis

The authors would like to express gratitude to the reviewer for taking the time to review this manuscript and for providing encouragement and helpful comments. All corrections made are marked in the revised manuscript for easier tracking by the reviewer.

Reviewer 2 Report

This is a very nicely written paper with many through details and well-organized information. The study aims to incorporate the structural benefits of using geosynthetic reinforcement to flexible pavement structure rutting performance of low-traffic roads. Please see below few minor comments. Thanks

Abstract: I recommend adding to the abstract an introductory / general sentence about the topic being discussed in the manuscript.

Page 2, line 27: state of Louisiana.

Please add more introductory information about geosynthetics in the first paragraph of your introduction.

Please separate the last paragraph of the introduction into an independent section entitled objectives.

In the description of Figure, a sentence should be added that geosynthetics were used in the upper portion of the subgrade and / or base. Please add some verbage to describe the red lines of Figure 1. I noticed that this is mentioned on page 12 line 207. If mentioned before, it avoid any potential confusion of the readers.

Page 16 line 259, please check if “equation 9” is the right equation you are referring to.

Can you please elaborate why the first 100 cycles are enough to calibrate the MEPDG transfer functions? Shouldn’t you be aiming for a higher number of cycles to guarantee a kind of secondary stage? I believe you are relying on the findings of a full-scale experiment but still I wanted to make the comment.

Please add couple of statements at the end of the “Summary and Conclusions” section elaborating on the implementation status of this study. Was there any project that incorporates geosynthetics and that was designed using the findings of this study?

Author Response

(The authors gave the same response as above.)
